# Drawing Robust Scratch Tickets: Subnetworks with Inborn Robustness Are Found within Randomly Initialized Networks

**Yonggan Fu[1], Qixuan Yu[1], Yang Zhang[2], Shang Wu[1], Xu Ouyang[1]**
**David Cox[2], Yingyan Lin[1]**
[1]Rice University, [2]MIT-IBM Watson AI Lab
{yf22,qy12,sw99,xo28}@rice.edu, {yang.zhang2,David.D.Cox}@ibm.com

## Abstract

Deep Neural Networks (DNNs) are known to be vulnerable to adversarial attacks, i.e., an imperceptible perturbation to the input can mislead DNNs trained on clean images into making erroneous predictions. To tackle this, adversarial training is currently the most effective defense method, by augmenting the training set with adversarial samples generated on the fly. **Interestingly, we discover for the first time that there exist subnetworks with inborn robustness, matching or surpassing the robust accuracy of the adversarially trained networks with comparable model sizes, within randomly initialized networks without any model training**, indicating that adversarial training on model weights is not indispensable towards adversarial robustness. We name such subnetworks Robust Scratch Tickets (RSTs), which are also by nature efficient. Distinct from the popular lottery ticket hypothesis, neither the original dense networks nor the identified RSTs need to be trained. To validate and understand this fascinating finding, we further conduct extensive experiments to study the existence and properties of RSTs under different models, datasets, sparsity patterns, and attacks, drawing insights regarding the relationship between DNNs' robustness and their initialization/overparameterization. Furthermore, we identify the poor adversarial transferability between RSTs of different sparsity ratios drawn from the same randomly initialized dense network, and propose a Random RST Switch (R2S) technique, which randomly switches between different RSTs, as a novel defense method built on top of RSTs. We believe our findings about RSTs have opened up a new perspective to study model robustness and extend the lottery ticket hypothesis. Our codes are available at: https://github.com/RICE-EIC/Robust-Scratch-Ticket.

## 1 Introduction

There has been an enormous interest in deploying deep neural networks (DNNs) into numerous real-world applications requiring strict security. Meanwhile, DNNs are vulnerable to adversarial attacks, i.e., an imperceptible perturbation to the input can mislead DNNs trained on clean images into making erroneous predictions. To enhance DNNs' robustness, adversarial training augmenting the training set with adversarial samples is commonly regarded as the most effective defense method. Nevertheless, adversarial training is time-consuming as each stochastic gradient descent (SGD) iteration requires multiple gradient computations to produce adversarial images. In fact, its actual slowdown factor over standard DNN training depends on the number of gradient steps used for adversarial example generation, which can result in a 3~30 times longer training time [1].

In this work, we ask an intriguing question: "*Can we find robust subnetworks within randomly initialized networks without any training*"? This question not only has meaningful practical implication

35th Conference on Neural Information Processing Systems (NeurIPS 2021).

but also potential theoretical ones. If the answer is yes, it will shed light on new methods towards robust DNNs, e.g., adversarial training might not be indispensable towards adversarial robustness; furthermore, the existence of such robust subnetworks will extend the recently discovered lottery ticket hypothesis (LTH) [2], which articulates that neural networks contain sparse subnetworks that can be effectively trained from scratch when their weights being reset to the original initialization, and thus our understanding towards robust DNNs. In particular, we make the following contributions:

- We discover for the first time that there exist subnetworks with inborn robustness, matching or surpassing the robust accuracy of adversarially trained networks with comparable model sizes, **within randomly initialized networks without any model training**. We name such subnetworks Robust Scratch Tickets (RSTs), which are also by nature efficient. Distinct from the popular LTH, neither the original dense networks nor the identified RSTs need to be trained. For example, RSTs identified from a randomly initialized ResNet101 achieve a 3.56%/4.31% and 1.22%/4.43% higher robust/natural accuracy than the adversarially trained *dense* ResNet18 with a comparable model size, under a perturbation strength of $\epsilon = 2$ and $\epsilon = 4$, respectively.

- We propose a general method to search for RSTs within randomly initialized networks, and conduct extensive experiments to identify and study the consistent existence and properties of RSTs under different DNN models, datasets, sparsity patterns, and attack methods, drawing insights regarding the relationship between DNNs' robustness and their initialization/overparameterization. Our findings on RSTs' existence and properties have opened up a new perspective for studying DNN robustness and can be viewed as a complement to LTH.

- We identify the poor adversarial transferability between RSTs of different sparsity ratios drawn from the same randomly initialized dense network, and propose a Random RST Switch (R2S) technique, which randomly switches between different RSTs, as a novel defense method built on top of RSTs. While an adversarially trained network shared among datasets suffers from degraded performance, our R2S enables the use of one random dense network for effective defense across different datasets.

## 2   Related Works

**Adversarial attack and defense.** DNNs are vulnerable to adversarial attacks, i.e., an imperceptible perturbation on the inputs can confuse the network into making a wrong prediction [3]. There has been a continuous war [4, 5] between adversaries and defenders. On the adversary side, stronger attacks continue to be proposed, including both white-box [6, 7, 8, 9, 10] and black-box ones [11, 12, 13, 14, 15], to notably degrade the accuracy of the target DNNs. In response, on the defender side, various defense methods have been proposed to improve DNNs' robustness against adversarial attacks. For example, randomized smoothing [16, 17] on the inputs can certifiably robustify DNNs against adversarial attacks; [18, 19, 20, 21, 22] purify the adversarial examples back to the distribution of clean ones; and [23, 24, 25] adopt detection models to distinguish adversarial examples from clean ones. In particular, adversarial training [26, 6, 27, 28] is currently the most effective defense method. In this work, we discover the existence of RST, i.e., there exist subnetworks with inborn robustness that are hidden in randomly initialized networks without any training.

**Model robustness and efficiency.** Both robustness and efficiency [29] matter for many DNN-powered intelligent applications. There have been pioneering works [30, 31, 32, 33, 34] that explore the potential of pruning on top of robust DNNs. In general, it is observed that moderate sparsity is necessary for maintaining DNNs' adversarial robustness, over-sparsified DNNs are more vulnerable [30], and over-parameterization is important for achieving decent robustness [6]. Pioneering examples include: [31] finds that naively pruning an adversarially trained model will result in notably degraded robust accuracy; [35] for the first time finds that quantization, if properly exploited, can even enhance quantized DNNs' robustness by a notable margin over their full-precision counterparts; and [32] adopts pruning techniques that are aware of the robust training objective via learning a binary mask on an adversarially pretrained network; and [34] prunes vulnerable features via a learnable mask and proposes a vulnerability suppression loss to minimize the feature-level vulnerability. In contrast, our work studies the existence, properties, and potential applications of RSTs, drawn from randomly initialized networks.

**Lottery ticket hypothesis.** The LTH [2] shows that there exist small subnetworks (i.e., lottery tickets (LTs)) within dense, randomly initialized networks, that can be trained alone to achieve comparable

accuracies to the latter. The following works aim to: (1) study LTs' properties [36, 37, 38, 39], (2) improve LTs' performance [40, 41, 42, 43], and (3) extend LTH to various networks/tasks/training-pipelines [44, 45, 46, 47, 48, 49, 50]. The LTs drawn by existing works need to be (1) drawn from a pretrained network (except [43]), and (2) trained in isolation after being drawn from the dense network. In contrast, RSTs are drawn from randomly initialized networks and, more importantly, no training is involved. We are inspired by [51], which finds that randomly weighted networks contain subnetworks with impressive accuracy and [52, 53], which provide theoretical support for [51] and extend [51] to identify binary neural networks in randomly weighted networks, respectively. While DNNs' adversarial robustness is important, it is not clear whether robust subnetworks exist within randomly initialized networks. We provide a positive answer and conduct a comprehensive study about the existence, properties, and potential applications for RSTs.

# 3 Robust Scratch Tickets: How to Find Them

In this section, we describe the method that we adopt to search for RSTs within randomly initialized networks. As modern DNNs contain a staggering number of possible subnetworks, it can be non-trivial to find subnetworks with inborn robustness in randomly weighted networks.

## 3.1 Inspirations from previous works

Our search method draws inspiration from prior works. In particular, [31] finds that naively pruning an adversarially trained model will notably degrade the robust accuracy, [32] learns a binary mask on an adversarially pretrained network to make pruning techniques aware of model robustness, and [51] shows that randomly weighted networks contain subnetworks with impressive nature accuracy, inspiring us to adopt learnable masks on top of randomly initialized weights to identify the RSTs.

## 3.2 The proposed search strategy

**Overview.** Our search method adopts a sparse and learnable mask $m$ associated with the weights of randomly initialized networks, where the search process for RSTs is equivalent to only update $m$ without changing the weights as inspired by [32, 51]. To search for practically useful RSTs, the update of $m$ has to (1) be aware of the robust training objective, and (2) ensure that the sparsity of $m$ is sufficiently high, e.g., higher than a specified value $(1 - k/N)$ where $k$ is the number of remaining weights and $N$ is the total number of weights. In particular, we adopt an adversarial search process to satisfy (1) and binarize $m$ to activate only a fraction of the weights in the forward pass to satisfy (2).

**Objective formulation.** We formulate the learning process of $m$ as a minimax problem:

$$\arg \min_{m} \sum_{i} \max_{\|\delta\|_\infty \leq \epsilon} \ell(f(\hat{m} \odot \theta, x_i + \delta), y_i) \quad s.t. \; ||\hat{m}||_0 \leqslant k \qquad (1)$$

where $\ell$ is the loss function, $f$ is a randomly initialized network with random weights $\theta \in \mathbb{R}^N$, $x_i$ and $y_i$ are the $i$-th input and label pair, $\odot$ denotes element-wise product operation, $\delta$ is a small perturbation applied to the inputs and $\epsilon$ is a scalar that limits the perturbation magnitude in terms of the $L_{inf}$ norm. Following [51], $\hat{m} \in \{0, 1\}^d$ approximates the top $k$ largest elements of $m \in \mathbb{R}^N$ using 1 and 0 otherwise during the forward pass to satisfy the target sparsity constraint, while all the elements in $m$ will be updated during the backward pass via straight-through estimation [54]. Such an adversarial search process guarantees the awareness of robustness, which differs from vanilla adversarial training methods [26, 6, 27, 28] in that here the model weights are never updated. After the adversarial search, a found RST is claimed according to the finally derived binary mask $\hat{m}$. As discussed in Sec. 4.5, $\hat{m}$ can have different sparsity patterns, e.g., element-wise, row-wise, or kernel-wise, where RSTs are consistently observed.

**Inner optimization.** We solve the inner optimization in Eq. 1 with Projected Gradient Descent (PGD) [6] which iteratively adopts the sign of one-step gradient as an approximation to update $\delta$ with a small step size $\alpha$, where the $t$-th iteration can be formulated as:

$$\delta_{t+1} = clip_\epsilon \{\delta_t + \alpha \cdot sign(\nabla_{\delta_t} \ell(f(\hat{m} \odot \theta, x_i + \delta_t), y_i))\} \qquad (2)$$

where $clip_\epsilon$ denotes the clipping function that enforces its input to the interval $[-\epsilon, \epsilon]$. As shown in Sec. 4.6, other adversarial training methods can also be adopted to draw RSTs.

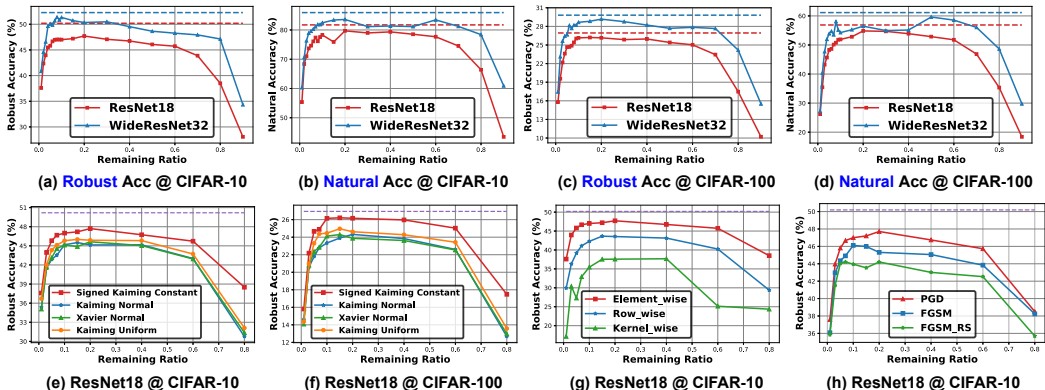

Figure 1: Illustrating RSTs' consistent existence, where (a)~(d): The robust and natural accuracy of RSTs with different remaining ratios in ResNet18 and WideRestNet32 on CIFAR-10/100, respectively; (e)~(f): The robust accuracy of RSTs with different remaining ratios identified in ResNet18 under different initialization methods on CIFAR-10/100, respectively; (g) The robust accuracy of RSTs with different sparsity patterns identified in ResNet18 on CIFAR-10; and (h) The robust accuracy of RSTs identified using different adversarial search methods in ResNet18 on CIFAR-10. The accuracies of the adversarially trained original dense networks are annotated using **dashed lines**.

## 4 The Existence of Robust Scratch Tickets

In this section, we validate the consistent existence of RSTs across different networks and datasets with various sparsity patterns and initialization methods based on the search strategy in Sec. 3.

### 4.1 Experiment Setup

**Networks and datasets.** Throughout this paper, we consider a total of four networks and three datasets, including PreActResNet18 (noted as ResNet18 for simplicity)/WideResNet32 on CIFAR-10/CIFAR-100 [55] and ResNet50/ResNet101 on ImageNet [56].

**Adversarial search settings.** For simplicity, we adopt layer-wise uniform sparsity when searching for RSTs, i.e., the weight remaining ratios $k/N$ for all the layers are the same. On CIFAR-10/CIFAR-100, we adopt PGD-7 (7-step PGD) training for the adversarial search based on Eq. 1 following [27, 6]. On ImageNet, we use FGSM with random starts (FGSM-RS) [27], which can be viewed as a 1-step PGD, to update the mask $m$. The detailed search settings can be found in the Appendix.

**Adversarial attack settings.** We adopt PGD-20 [6] attacks with $\epsilon = 8$, which is one of the most effective attacks, if not specifically stated. We further examine the robustness of RSTs under more attack methods and larger perturbation strengths in Sec. 5.3 and Sec. 5.5, respectively.

**Model initialization.** As RSTs inherit the weights from randomly initialized networks, initialization can be crucial for their robustness. We consider four initialization methods: the Signed Kaiming Constant [51], Kaiming Normal [57], Xavier Normal [58], and Kaiming Uniform [57]. If not specifically stated, we adopt the Signed Kaiming Constant initialization thanks to its most competitive results (see Sec. 4.4). We report the average results based on three runs with different random seeds.

### 4.2 RSTs exist on CIFAR-10/100

Fig. 1 (a)~(d) visualize both the robust and natural accuracies of the identified RSTs with different weight remaining ratios (i.e., $k/N$) in two networks featuring different degrees of overparameterization on CIFAR-10/100. We also visualize the accuracy of the original dense networks *after adversarial training*, while RSTs are drawn from the same dense networks with only random initialization.

**Observations and analysis.** RSTs do exist as the drawn RSTs with a wide range of weight remaining ratios achieve both a decent robust and natural accuracy, even without any model training. Specifically, we can see that (1) RSTs can achieve a comparable robust and natural accuracy with the adversarially trained original dense networks under a wide range of remaining ratios (i.e., 5%~40%), e.g., the RST drawn from ResNet18 with a remaining ratio of 20% (i.e., 80% sparsity) without any training suffers from only 2.08% and 2.43% drop in the robust and natural accuracy, respectively, compared with the adversarially trained dense ResNet18; (2) RSTs hidden within randomly initialized networks

with more overparameterization achieve a better robustness than the adversarially trained *dense* networks with a similar model size, e.g., the RST with a remaining ratio of 10% identified in the randomly initialized WideResNet32 achieves a 1.14%/1.72% higher robust/natural accuracy with a 60% reduction in the total number of parameters without any training, over the adversarially trained *dense* ResNet18; and (3) RSTs under very large or small remaining ratios have relatively inferior robust and natural accuracies, which is expected since RSTs under large remaining ratios (e.g., > 60%) are close to the original randomly initialized networks that mostly make random predictions, while RSTs under small weight remaining ratios (e.g., 5%) are severely underparameterized.

### 4.3 RSTs exist on ImageNet

We further validate the existence of RSTs on ImageNet. As shown in Tab. 1, we can see that (1) RSTs do exist on ImageNet, e.g., RSTs identified within ResNet101 achieve a 3.56%/4/31% and 1.22%/4.43% higher robust/natural accuracy than the adversarially trained ResNet18 with a comparable model size, under

Table 1: Comparing RSTs identified in ResNet50/101 with the adversarially trained ResNet18 on ImageNet.

| Network | Remaining Ratio | $\epsilon$ | # Parameter (M) | Natural Acc (%) | Robust Acc (%) |
|---------|-----------------|------------|-----------------|-----------------|----------------|
| ResNet18 (Dense) | - | 2 | 11.17 | 56.25 | 37.09 |
| RST @ ResNet50 | 0.4 | 2 | 9.41 | 53.84 | 35.24 |
| RST @ ResNet101 | 0.25 | 2 | 10.62 | **60.56** | **40.65** |
| ResNet18 (Dense) | - | 4 | 11.17 | 50.41 | 25.46 |
| RST @ ResNet50 | 0.4 | 4 | 9.41 | 46.36 | 21.99 |
| RST @ ResNet101 | 0.25 | 4 | 10.62 | **54.84** | **26.68** |

$\epsilon = 2$ and 4, respectively; and (2) RSTs identified from more overparameterized networks achieve a better robust/natural accuracy than those from lighter networks, as RSTs from ResNet101 are notably more robust than those from ResNet50 under comparable model sizes.

### 4.4 RSTs exist under different initialization methods

We mainly consider Signed Kaiming Constant initialization [51] for the randomly initialized networks, as mentioned in Sec. 4.1, and here we study the influence of different initialization methods on RSTs. Fig. 1 (e) and (f) compare the robust accuracy of the RSTs drawn from ResNet18 with different initializations on CIFAR-10/100, respectively. Here we only show the robust accuracy for better visual clarity, as the corresponding natural accuracy ranking is the same and provided in the Appendix.

**Observations and analysis.** We can see that (1) RSTs consistently exist, when using all the initialization methods; and (2) RSTs drawn under the Signed Kaiming Constant [51] initialization consistently achieve a better robustness than that of the other initialization methods under the same weight remaining ratios, indicating that RSTs can potentially achieve stronger robustness with more dedicated initialization. We adopt Signed Kaiming Constant initialization in all the following experiments.

### 4.5 RSTs exist under different sparsity patterns

We mainly search for RSTs considering the commonly used element-wise sparsity, as mentioned in Sec. 4.1. A natural question is whether RSTs exist when considering other sparsity patterns, and thus here we provide an ablation study. Specifically, for each convolutional filter, we consider another three sparsity patterns: row-wise sparsity, kernel-wise sparsity, and channel-wise sparsity.

**Observations and analysis.** Fig. 1 (g) shows that RSTs with more structured row-wise and kernel-wise sparsity still exist, although suffering from a larger robustness accuracy drop over the adversarially trained original dense networks, and we fail to find RSTs with a decent robustness when considering channel-wise sparsity. Indeed, RSTs with more structured sparsity show a more inferior accuracy, since it is less likely that a consecutive structure within a randomly initialized network will be lucky enough to extract meaningful and robust features. Potentially, customized initialization can benefit RSTs with more structured sparsity patterns and we leave this as a future work.

### 4.6 RSTs exist under different adversarial search methods

To study RSTs' dependency on the adopted adversarial search methods, here we adopt another two adversarial search schemes, FGSM [3] and FGSM-RS [27], for the inner optimization in Eq. 1.

**Observations and analysis.** Fig. 1 (h) shows that (1) using an FGSM or FGSM-RS search scheme can also successfully identify RSTs from randomly initialized networks; and (2) RSTs identified uing PGD-7 training consistently show a better robustness under the same weight remaining ratios compared with the other two variants. This indicates that better adversarial search schemes can lead to higher quality RSTs and there potentially exist more robust RSTs within randomly initialized networks than the reported ones, when adopting more advanced adversarial search schemes.

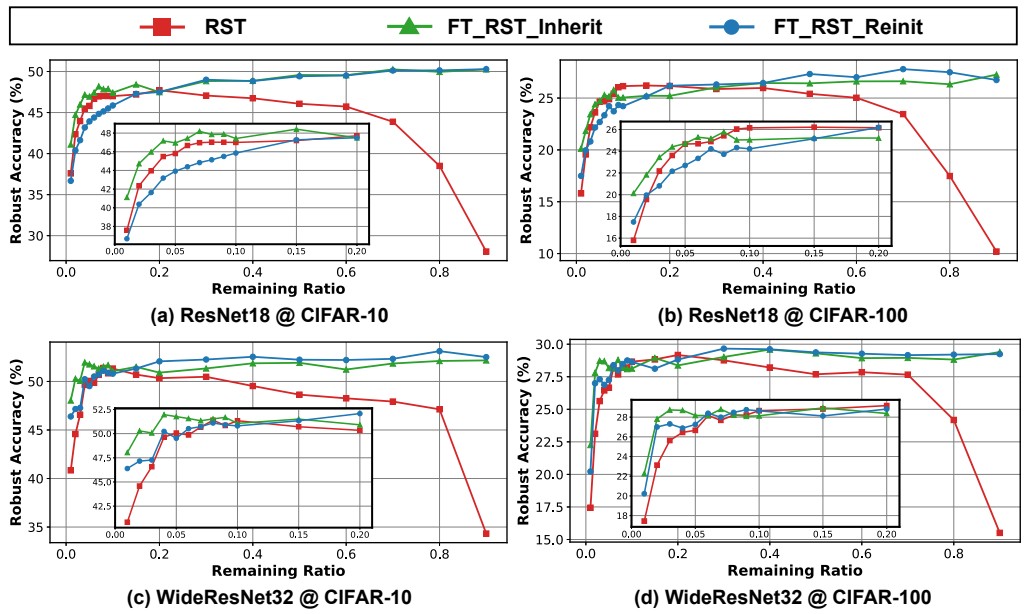

Figure 2: Comparing the robust accuracy of RSTs, fine-tuned RSTs with inherited weights, and fine-tuned RSTs with reinitialization, with zoom-ins for the low remaining ratios (1%~20%).

### 4.7 Comparison with other adversarial lottery tickets

We further benchmark RSTs with other adversarial lottery tickets identified by [59]. On top of WideResNet32 on CIFAR-10, [59] achieves a 50.48% robust accuracy with a sparsity of 80% under PGD-20 attacks with $\epsilon = 8$, while our RST achieves a 51.39% robust accuracy with a sparsity of 90%, which indicates that RSTs are high quality subnetworks with decent robustness even *without model training*. As will be shown in Sec. 5.1, the robustness of RSTs can be further improved after being adversarially fine-tuned.

### 4.8 Insights behind RSTs

The robustness of RSTs is attributed to RSTs' searching process which ensures the searched subnetworks to effectively identify critical weight locations for model robustness. This has been motivated by [60] that the location of weights holds most of the information encoded by the training, indicating that searching for the locations of a subset of weights within a randomly initialized network might be potentially as effective as adversarially training the weight values, in terms of generating robust models. In another word, training the model architectures may be an orthogonal and effective complement for training the model weights.

## 5 The Properties of Robust Scratch Tickets

In this section, we systematically study the properties of the identified RSTs for better understanding.

### 5.1 Robustness of vanilla RSTs vs. fine-tuned RSTs

An interesting question is "*how do the vanilla RSTs perform as compared to RSTs with adversarially fine-tuned model weights*"? Inspired by [2], we consider two settings: (1) fine-tuned RSTs starting with model weights inherited from the vanilla RSTs, and (2) fine-tuned RSTs with reinitialized weights. Fig. 2 compares their accuracy on three networks and two datasets with zoom-ins under low remaining ratios (1%~20%). Here we only show the robust accuracy for better visual clarity, where the natural accuracy with a consistent trend is provided in the Appendix.

**Observations and analysis.** We can see that (1) the vanilla RSTs achieve a comparable robust accuracy with that of the fine-tuned RSTs under a wide range of weight remaining ratios according to Fig. 2 (a), excepting under extremely large and small remaining ratios for the same reason as analyzed in Sec. 4.2; (2) under low remaining ratios (1%~20%), fine-tuned RSTs with inherited weights can mostly achieve the best robustness among the three RST variants, but the robust accuracy gap with

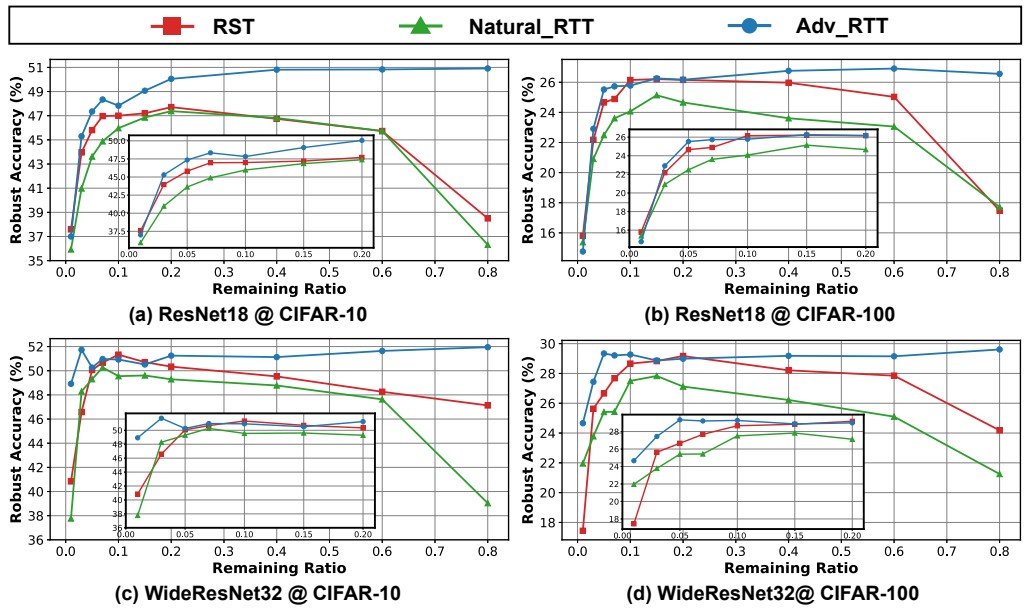

Figure 3: The robust accuracy achieved by RSTs, natural RTTs, and adversarial RTTs drawn from ResNet18/WideResNet32 on CIFAR-10/100, with zoom-ins under low remaining ratios (1%~20%).

the vanilla untrained RSTs is within -0.22%~3.52%; and (3) under commonly used weight remaining ratios (5%~20%) [61], fine-tuned RSTs with re-initialization can only achieve a comparable or even inferior robust accuracy compared with the vanilla RSTs without any model training. We also find that under extremely low remaining ratios like 1% (i.e., 99% sparsity), fine-tuned RSTs with re-initialization achieve a better robustness than the vanilla RSTs on extremely overparameterized networks (e.g, WideResNet32) since they still retain enough capacity for effective feature learning.

**Key insights.** The above observations indicate that the existence of RSTs under low weight remaining ratios reveals **another lottery ticket phenomenon** in that (1) an RST without any model training can achieve a better or comparable robust and natural accuracy with an adversarially trained subnetwork of the same structure, and (2) the fine-tuned RSTs with inherited weights can achieve a notably better robust accuracy than the re-initialized ones, indicating that RSTs win a better initialization.

## 5.2 Vanilla RSTs vs. RTTs drawn from trained networks

As the adversarial search in Eq. 1 can also be applied to trained networks, here we study whether subnetworks searched from trained models, called robust trained tickets (RTTs), can achieve a comparable accuracy over the vanilla RSTs. Fig. 3 compares RST with two kinds of RTTs, i.e., natural RTTs and adversarial RTTs which are searched from naturally trained and adversarially trained networks, respectively. For better visual clarity, we only show the robust accuracy in Fig. 3, and the corresponding natural accuracy with a consistent trend is provided in the Appendix.

**Observations and analysis.** We can observe that (1) adversarial RTTs consistently achieve the best robustness, indicating that under the same weight remaining ratio, subnetworks drawn from adversarially trained networks are more robust than RSTs drawn from randomly initialized ones or RTTs from naturally trained ones; (2) there exist natural RTTs with decent robustness even if the robust accuracy of the corresponding trained dense networks is almost zero; and (3) interestingly, RSTs consistently achieve a better robust/natural accuracy over natural RTTs, indicating the subnetworks from naturally trained networks are less robust than the ones drawn from randomly initialized neural networks.

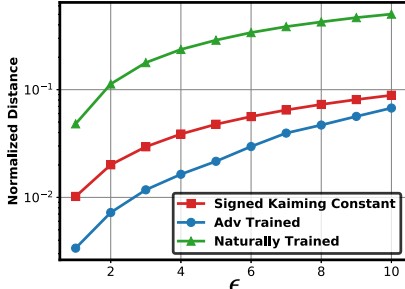

Figure 4: Normalized distances between the feature maps generated by clean and noisy images on ResNet18 / CIFAR-10.

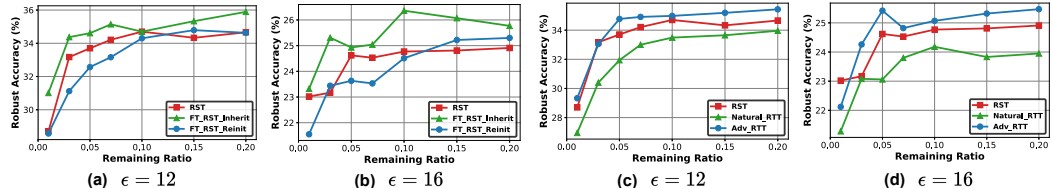

|  (a) $\epsilon = 12$ | (b) $\epsilon = 16$ | (c) $\epsilon = 12$ | (d) $\epsilon = 16$ |

Figure 6: Validating RSTs' properties under stronger perturbations on ResNet18 / CIFAR-10. (a)∼(b): Comparing vanilla RSTs with fine-tuned RSTs under $\epsilon$ =12 and 16, respectively; and (c)∼(d): Comparing vanilla RSTs with RTTs under $\epsilon$ =12 and 16, respectively.

To understand the above observations, we visualize the normalized distance between the feature maps of the last convolution layer generated by (1) clean images and (2) noisy images (i.e., clean images plus a random noise with a magnitude of $\epsilon$ on top of ResNet18 with random initialization (Signed Kaiming Constant [51]), weights trained on clean images, and weights trained on adversarial images on CIFAR-10), inspired by [62]. Fig. 4 shows that the naturally trained ResNet18 suffers from large distance gaps between the feature maps with clean and noisy inputs, indicating that it is more sensitive to the perturbations applied to its inputs and thus the natural RTTs identified within it are potentially less robust, while the adversarially trained ResNet18 is the opposite.

**Is overparameterization necessary in adversarial training?** We further compare the vanilla RSTs, fine-tuned RSTs with inherited weights, and adversarial RTTs, drawn from ResNet18 on CIFAR-10 in Fig. 5. We can see that adversarial RTTs consistently achieve the best robustness under the same weight remaining ratio. The comparison between adversarial RTTs and fine-tuned RSTs with inherited weights indicates that robust subnetworks within an adversarially trained and overparameterized network can achieve a better robustness than those drawn from a randomly initialized network or adversarially trained from scratch, i.e., the overparameterization during adversarial training is necessary towards decent model robustness.

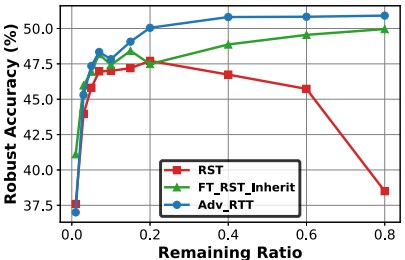

Figure 5: Robust accuracy vs. remaining ratio for RSTs, fine-tuned RSTs with inherited weights, and adversarial RTTs.

### 5.3 RSTs' scalability to stronger perturbations

To explore whether the findings in Sec. 5.1 and Sec. 5.2 can be scaled to more aggressive attacks with stronger perturbations, we visualize the comparison between RSTs and fine-tuned RSTs/RTTs in Fig. 6, when considering the perturbation strength $\epsilon = 12, 16$ in ResNet18 on CIFAR-10.

**Observations and analysis.** We can observe that (1) RSTs still can be successfully identified under stronger perturbations; and (2) the insights in both Sec. 5.1 and Sec. 5.2 still hold and the accuracy gaps between the vanilla RSTs and natural RTTs are even larger under stronger perturbations.

### 5.4 RSTs' robustness against $L_2$-PGD attacks

We further evaluate RSTs with different remaining ratios and their corresponding dense networks trained by $L_{inf}$-PGD against $L_2$-PGD attacks on top of ResNet18/WideResNet32 on CIFAR-10. As shown in Tab. 2, we can see that with increased perturbation strength $\epsilon$, the dense networks suffer from larger robust accuracy drops and RSTs can gradually outperform the dense networks by a notable margin, indicating that RSTs will suffer less from overfitting to a specific norm-based attack than the dense networks.

Table 2: Evaluating RSTs and the corresponding dense networks trained by $L_{inf}$-PGD against $L_2$-PGD attacks with different perturbation strengths on CIFAR-10.

| Network | ResNet18 | | | WideResNet32 | | |
|---|---|---|---|---|---|---|
| | $\epsilon$=0.50 | $\epsilon$=1.71 | $\epsilon$=2.00 | $\epsilon$=0.50 | $\epsilon$=1.71 | $\epsilon$=2.00 |
| Dense | 63.61 | 40.85 | 39.04 | 63.94 | 36.55 | 34.66 |
| RST@5% | 61.11 | 43.81 | 42.25 | 65.96 | 47.75 | 45.96 |
| RST@7% | 63.04 | 43.94 | 42.41 | 65.05 | 43.63 | 42.08 |
| RST@10% | 63.61 | 44.49 | 42.81 | 66.93 | 45.56 | 43.87 |
| RST@15% | 62.03 | 44.16 | 42.58 | 67.18 | 45.23 | 43.44 |
| RST@20% | 64.43 | 44.35 | 42.51 | 67.54 | 45.21 | 43.51. |

### 5.5 RSTs' robustness against more adversarial attacks

We also evaluate the robustness of RSTs against more adversarial attacks, including the CW-L2/CW-Inf attacks [8], an adaptive attack called Auto-Attack [7], and the gradient-free Bandits [12] as

detailed in the Appendix. We find that RSTs are generally robust against different attacks, e.g., under Auto-Attack/CW-Inf attacks on CIFAR-10, the RST with a weight remaining ratio of 5% identified in a randomly initialized WideResNet32 achieves a 2.51%/2.80% higher robust accuracy, respectively, together with an 80% parameter reduction as compared to the adversarially trained ResNet18.

# 6 Applications of the Robust Scratch Tickets

In this section, we show that RSTs can be utilized to build a defense method, serving as the first heuristic to leverage the existence of RSTs for practical applications. Specifically, we first identify the poor adversarial transferability between RSTs of different weight remaining ratios drawn from the same randomly initialized networks in Sec. 6.1, and then propose a technique called R2S in Sec. 6.2. Finally, we evaluate R2S's robustness over adversarially trained networks in Sec. 6.3.

## 6.1 Poor adversarial transferability between RSTs with different weight remaining ratios

To study the adversarial transferability between RSTs of different remaining ratios drawn from the same randomly initialized networks, we transfer the adversarial attacks generated under one remaining ratio to attack RSTs with other remaining ratios and visualize the resulting robust accuracy on ResNet18 with CIFAR-10 in Fig. 7. We can see that the robust accuracies in the diagonal (i.e., the same RST for generating attacks and inference) are notably lower than those of the non-diagonal ones, i.e., the transferred ones. Similar observations, provided in the Appendix, can consistently be found on different networks/datasets, indicating that the adversarial transferability between RSTs is poor and thus providing potential opportunities for developing new defense methods.

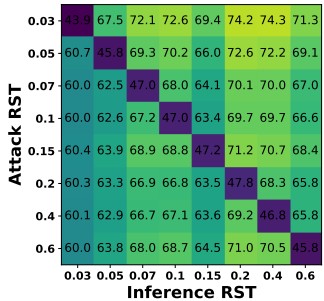

Figure 7: Adversarial transferability between RSTs from the same randomly initialized ResNet18 on CIFAR-10, where the robust accuracy is annotated.

## 6.2 The proposed R2S technique

RSTs have two attractive properties: (1) RSTs with different remaining ratios from the same networks inherently have a weight-sharing nature; and (2) the adversarial transferability between RSTs of different remaining ratios is poor (see Fig. 7). To this end, we propose a simple yet effective technique called R2S to boost model robustness. Specifically, R2S randomly switches between different RSTs from a candidate RST set during inference, leading to a mismatch between the RSTs adopted by the adversaries to generate attacks and the RSTs used for inference. In this way, robustness is guaranteed by the poor adversarial transferability between RSTs while maintaining parameter-efficiency thanks to the weight-sharing nature of RSTs with different weight remaining ratios drawn from the same networks. The only overhead is the required binary masks for different RSTs, which is negligible (e.g., 3%) as compared with the total model parameters. In addition, as evaluated in Sec. 6.3, the same randomly initialized network can be shared among *different tasks* with task-specific RSTs, i.e., only extra binary masks are required for more tasks, further improving parameter efficiency.

Table 3: Evaluating R2S under PGD-20 on CIFAR-10 (columns 2~3) and CIFAR-100 (columns 4~5).

| Network | Natural Acc (%) | PGD-20 Acc (%) | Natural Acc (%) | PGD-20 Acc (%) |
|---|---|---|---|---|
| ResNet18 (Dense) | 81.73 | 50.19 | 56.88 | 26.94 |
| R2S @ ResNet18 | 77.49 | **63.93** | 52.12 | **41.01** |
| WideResNet32 (Dense) | 85.93 | 52.27 | 61.14 | 29.81 |
| R2S @ WideResNet32 | 81.87 | **67.55** | 54.72 | **43.27** |

## 6.3 Evaluating the R2S technique

**Evaluation setup.** Since both the adversaries and defenders could adjust the probability for their RST choices, we assume that both adopt uniform sampling from the same candidate RST set for simplicity. We adopt an RST candidate set with remaining ratios [5%, 7%, 10%, 15%, 20%, 30%] and results under different RST choices can be found in the Appendix.

Table 4: Evaluating R2S under Auto-Attack [7] and CW-Inf attack [8] on CIFAR-10.

| Network | Auto-Attack $\epsilon = 8$ | Auto-Attack $\epsilon = 12$ | CW-Inf $\epsilon = 8$ | CW-Inf $\epsilon = 12$ |
|---|---|---|---|---|
| ResNet18 (Dense) | 46.39 | 41.13 | 48.99 | 45.91 |
| R2S @ ResNet18 | 51.92 | **49.77** | 54.02 | **52.63** |
| WideResNet32 (Dense) | 49.66 | 44.97 | 54.16 | 49.22 |
| R2S @ WideResNet32 | 56.06 | **53.91** | 58.55 | **57.48** |

**R2S against SOTA attacks.** As shown in Tabs. 3 and 4, we can see that R2S consistently boosts the robust accuracy under all the networks/datasets/attacks, e.g., a 13.74% ~ 15.28% and 5.53% ~ 8.94% higher robust accuracy under PGD-20 and Auto-Attack, respectively, on CIFAR-10.

**R2S against adaptive attacks.** We further design two adaptive attacks on top of PGD-20 attacks for evaluating our R2S: (1) *Expectation over Transformation (EOT)* following [63], which generates adversarial examples via the expectations of the gradients from all candidate RSTs; and (2) *Ensemble*, which generates attacks based on the ensemble of all the candidate RSTs, whose prediction is the averaged results of all candidate RSTs. Both attacks consider the information of all candidate RSTs. As shown in Tab. 5, we can observe that the robust accuracy of R2S against the two adaptive attacks will drop compared with our R2S against vanilla PGD-20 attacks in Tab. 3, yet it's still 7.40%~9.83% higher than the adversarially trained dense networks, indicating that the two adaptive attacks are effective adaptive attacks and our R2S can still be a strong technique to boost adversarial robustness.

Table 5: Robust accuracy of R2S against two adaptive attacks on CIFAR-10 (columns 2~3) and CIFAR-100 (columns 4~5).

| Method | ResNet-18 | WideResNet-32 | ResNet-18 | WideResNet-32 |
|---|---|---|---|---|
| Dense@PGD-20 | 50.19 | 52.27 | 26.94 | 29.81 |
| R2S@*EOT* | **57.59** | **64.98** | **36.68** | **40.17** |
| R2S@*Ensemble* | **59.05** | **62.1** | **35.98** | **38.25** |

**Applying random switch on adversarial RTTs.** Indeed, the R2S technique can also be extended to adversarial RTTs as the poor adversarial transferability still holds between adversarial RTTs with different remaining ratios (see the Appendix). For example, when adopting the same candidate RST sets as in Tab. 3, R2S on adversarial RTTs achieves a 0.84%/1.58% higher robust/natural accuracy over R2S using RSTs on CIFAR-10 under PGD-20 attacks, indicating that R2S is a general technique to utilize robust tickets. More results of R2S using adversarial RTTs are in the Appendix.

**Advantages of RSTs over adversarial RTTs.** A unique property of RSTs is that the same randomly initialized network can be shared among different tasks and task-specific RSTs can be drawn from this same network and stored compactly via instantiated binary masks, leading to advantageous parameter efficiency. In contrast, the task-specific adversarial RTTs drawn from adversarially trained networks on a different task will lead to a reduced robustness.

To evaluate this, we search for the CIFAR-100-specific adversarial RTTs from a ResNet18 adversarially trained on CIFAR-10 (with the last fully-connected layer reinitialized), denoted as transferred adversarial RTTs, and compare it with (1) CIFAR-100-specific RSTs and (2) CIFAR-100-specific adversarial RTTs (identified from ResNet18 trained on CIFAR-100). As shown in Fig. 8, we can see that the transferred adversarial RTTs consistently achieve the worst robust accuracy, e.g., a 1.68% robust accuracy drop over the corresponding RST under a remaining ratio of 5%. Therefore, it is a unique advantage of RSTs to improve parameter efficiency, especially when more tasks are considered under resource constrained applications, while simultaneously maintaining decent robustness.

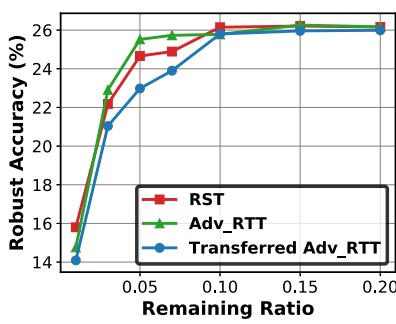

Figure 8: Comparing transferred adversarial RTTs with RSTs and adversarial RTTs in ResNet18 on CIFAR-100.

## 7 Conclusion

In this work, we show for the first time that there exist subnetworks with inborn robustness, matching or surpassing the robust accuracy of the adversarially trained networks with comparable model sizes, within randomly initialized networks without any model training. Distinct from the popular LTH, neither the original dense networks nor the identified RSTs need to be trained. To validate and understand our RST finding, we further conduct extensive experiments to study the existence and properties of RSTs under different models, datasets, sparsity patterns, and attacks. Finally, we make a heuristic step towards the practical uses of RSTs. In particular, we identify the poor adversarial transferability between RSTs of different sparsity ratios drawn from the same randomly initialized dense network, and propose a novel defense method called R2S, which randomly switches between different RSTs. While an adversarially trained network shared among datasets suffers from degraded performance, R2S enables one random dense network for effective defense across different datasets.

## Acknowledgements

The work is supported by the National Science Foundation (NSF) through the MLWiNS program (Award number: 2003137).

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
