# Supplementary Materials of Drawing Robust Scratch Tickets: Subnetworks with Inborn Robustness Are Found within Randomly Initialized Networks

**Yonggan Fu[1], Qixuan Yu[1], Yang Zhang[2], Shang Wu[1], Xu Ouyang[1]**
**David Cox[2], Yingyan Lin[1]**
[1]Rice University, [2]MIT-IBM Watson AI Lab
{yf22,qy12,sw99,xo28}@rice.edu, {yang.zhang2,David.D.Cox}@ibm.com

## 1 Overview and Outline

In this supplement, we provide more analysis about (1) the properties of RSTs, (2) the applications of R2S on top of RSTs/RTTs, and (3) the experiment details, providing a complement to the main content as outlined below:

- We provide the detailed search and attack settings in Sec. 2;

- We evaluate RSTs' robustness against more adversarial attacks in Sec. 3;

- We provide the natural accuracy achieved by RSTs, fine-tuned RSTs, and RTTs in Sec. 4;

- We provide more discussions about the adversarial transferability between RSTs and evaluate R2S with more candidate RST sets in Sec. 5 and Sec. 6, respectively;

- We visualize the adversarial transferability between adversarial RTTs and apply R2S on top of them in Sec. 7 and Sec. 8, respectively;

- We further demonstrate the advantages of RSTs over adversarial RTTs in Sec. 9.

## 2 Detailed Search and Attack Settings

**Adversarial search settings on CIFAR-10/100.** We adopt PGD-7 (7-step PGD) training for the adversarial search and update the mask $m$ for a total of 160 epochs using an SGD optimizer with a momentum of 0.9 and an initial learning rate of 0.1 which decays by 0.1 at both the 80-th and 120-th epochs with a batch size of 256.

**Adversarial search settings on ImageNet.** We adopt FGSM with random starts (FGSM-RS) [1] for the adversarial search and update the mask $m$ for a total of 100 epochs using an SGD optimizer with a momentum of 0.9 and a cosine learning rate decay with an initial learning rate of 0.256 and a batch size of 256 following [2].

**Attack settings.** We evaluate the robustness of RSTs under (1) PGD-20 attacks [3], (2) CW-L2/CW-Inf attacks [4], (3) Auto-Attack [5], and (4) the gradient-free attack Bandits [6]. We mainly evaluate RSTs' robustness under PGD-20 attacks in the main text and show RSTs' consistent robustness against other attacks in Sec. 3. In particular, for the CW-L2/CW-Inf attacks we adopt the implementation in AdverTorch [7] and follow [8, 9] to adopt 1 search step on $c$, which balances the distance constraint and the objective function, for 100 iterations with an initial constant of 0.1 and a learning rate of 0.01; for the Auto-Attack [5] and Bandits [6], we adopt the official implementation and default settings in their original papers.

35th Conference on Neural Information Processing Systems (NeurIPS 2021).

Table 1: Evaluating the robustness of RSTs with different weight remaining ratios against the Auto-Attack [5], CW-L2/CW-Inf attacks [4], and Bandits [6] on top of two networks on CIFAR-10.

| Network | Remaining Ratio (%) | Auto-Attack | | CW-L2 | CW-Inf | | Bandits | |
|---|---|---|---|---|---|---|---|---|
| | | $\epsilon$=8 | $\epsilon$=12 | - | $\epsilon$=8 | $\epsilon$=12 | $\epsilon$=8 | $\epsilon$=12 |
| ResNet18 (Dense) | 100% | 46.39 | 41.13 | 52.78 | 48.99 | 45.91 | 66.99 | 64.17 |
| RST @ ResNet18 | 5% | 45.75 | 41.43 | 51.74 | 48.07 | 45.57 | 61.18 | 59.30 |
| | 10% | 45.30 | 40.84 | 52.03 | 47.95 | 45.05 | 63.82 | 61.55 |
| | 20% | 42.55 | 37.97 | 49.35 | 45.03 | 42.45 | 65.47 | 62.80 |
| | 30% | 45.14 | 39.94 | 51.97 | 47.76 | 44.08 | 65.03 | 62.40 |
| | 50% | 42.19 | 37.75 | 49.07 | 44.85 | 42.27 | 64.10 | 61.43 |
| | 70% | 42.18 | 37.80 | 48.14 | 44.44 | 42.18 | 60.17 | 57.97 |
| WideResNet32 (Dense) | 100% | 49.66 | 44.97 | 57.13 | 54.16 | 49.22 | 69.28 | 65.86 |
| RST @ WideResNet32 | 5% | 48.90 | 44.76 | 55.58 | 51.53 | 48.79 | 66.04 | 64.00 |
| | 10% | 43.91 | 43.67 | 55.67 | 51.47 | 48.38 | 67.97 | 65.12 |
| | 20% | 46.80 | 43.75 | 55.33 | 51.25 | 48.17 | 68.54 | 65.66 |
| | 30% | 48.91 | 44.53 | 57.38 | 52.95 | 49.60 | 67.75 | 65.11 |
| | 50% | 48.28 | 42.35 | 55.93 | 50.74 | 47.48 | 66.80 | 64.22 |
| | 70% | 44.46 | 39.04 | 51.30 | 47.33 | 44.36 | 67.08 | 64.56 |

**Evaluation settings for R2S.** We assume that adversaries generate adversarial examples based on a randomly selected RST from a candidate RST set and our R2S then randomly selects an RST from the same set for performing inference. Such assumption does not lose the generality since (1) any RST out of the candidate RST set selected by the adversaries will merely increase the achieved robust accuracy of R2S due to the mismatch between the RSTs for inference and generating attacks, (2) while adversaries may tend to select RSTs with better attacking success rates, our R2S can also increase the probability of sampling more robust RSTs for stronger defense, and (3) the average robust accuracy under each attack RST (i.e., the RST with a specific weight remaining ratio for generating adversarial examples) is similar, thus the random selection strategy can be a good approximation of the defense effectiveness. Here we assume both the adversaries and R2S adopt the same candidate RST set for simplicity. In particular, we adopt a candidate RST set of [5%, 7%, 10%, 15%, 20%, 30%] in Sec. 6.3 of the main text, and the robustness under other set options can be found in Sec. 6.

## 3 RSTs' Robustness against More Adversarial Attacks

We evaluate the identified RSTs' robustness against more attacks on top of two networks on CIFAR-10 as a complement for Sec. 5.4 in the main text. As observed from Tab. 1, we can see that the RSTs searched by PGD-7 training are also robust against other attacks. For example, under the Auto-Attack with $\epsilon$ =8/12, the RST with a remaining ratio of 5% in WideResNet32 achieves (1) only a 0.76%/0.21% drop in robust accuracy together with a 95% reduction in the model size, as compared to the adversarially trained dense WideResNet32, and (2) a 2.51%/3.63% higher robust accuracy and a 80% reduction in the model size, compared with the adversarially trained dense ResNet18. This set of experiments indicates that the identified RSTs are generally robust without overfitting to one specific type of attacks.

## 4 The Natural Accuracy of RSTs, Fine-tuned RSTs, and RTTs

Here we provide the natural accuracy of RSTs and their variants as a complement to the the robust accuracy shown in Sec. 4 and Sec. 5 of the main text.

**The natural accuracy of RSTs under different initializations.** As shown in Fig. 1, RSTs under Signed Kaiming Constant initialization consistently achieve the best natural accuracy and the overall ranking between different initialization methods is generally consistent with the robust accuracy ranking in Sec. 4.4 of the main text.

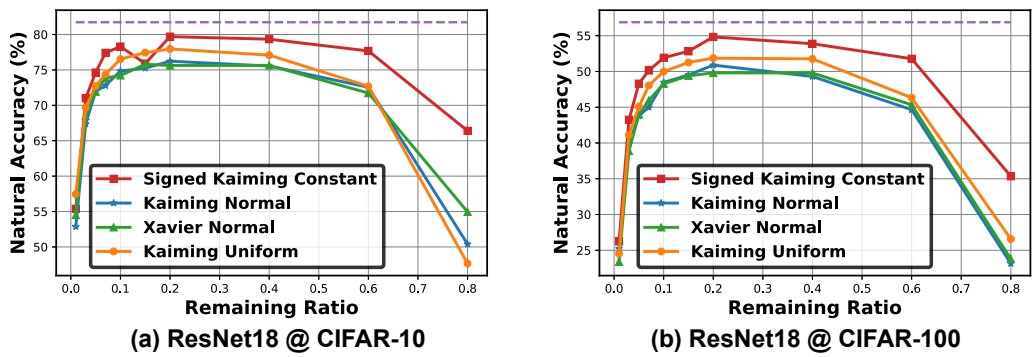

Figure 1: The natural accuracy of RSTs identified from different initializations under different weight remaining ratios and found in ResNet18 on (a) CIFAR-10 and (b) CIFAR-100.

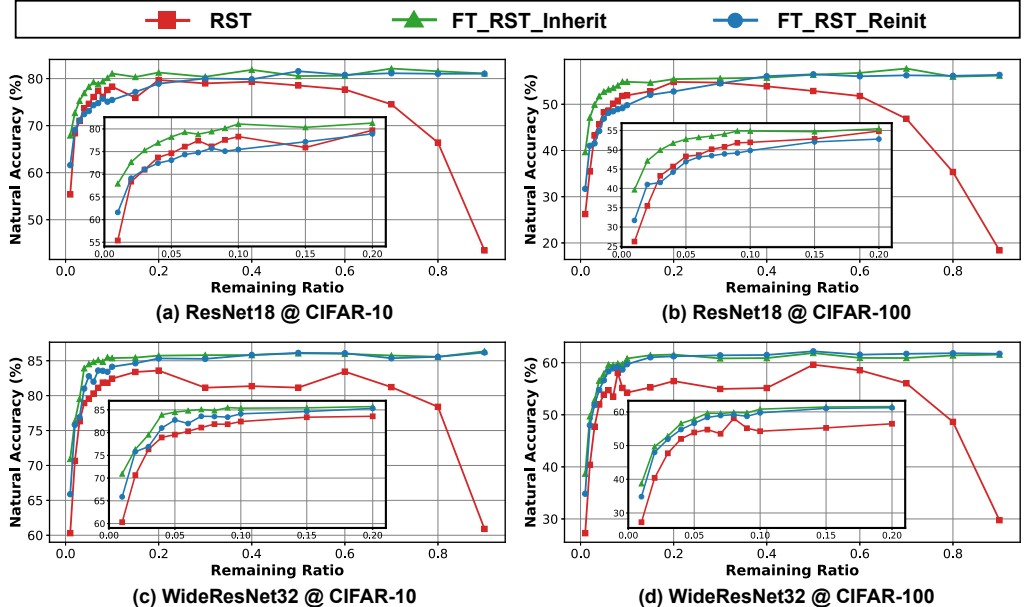

Figure 2: Comparing the natural accuracy of RSTs, fine-tuned RSTs with inherited weights, and fine-tuned RSTs with reinitialization, with zoom-ins for the low remaining ratios (1%~20%).

**The natural accuracy of RSTs and fine-tuned RSTs.** As observed from Fig. 2, we can see that our analysis about the lottery ticket phenomenon in Sec. 5.1 of the main text still holds for the natural accuracy. In particular, (1) the vanilla RSTs can generally achieve a comparable natural accuracy with that of the fine-tuned RSTs, indicating that the untrained RSTs naturally achieve a good balance between their robust and natural accuracy according to both Fig. 2 here and Fig. 2 in the main text; (2) under low weight remaining ratios (1%~20%), fine-tuned RSTs with inherited weights mostly achieve the best natural accuracy among the three RST variants, indicating that RSTs win better initializations; and (3) the vanilla RSTs on ResNet18 without any training can achieve a comparable natural accuracy compared with fine-tuned RSTs with re-initialization, while the latter ones on WideResNet32 can achieve a slightly better natural accuracy, which we conjecture is resulting from a higher degree of overparameterization in the original networks and thus indicates that the drawn RSTs favor a stronger capacity for effective feature learning during the fine-tuning process.

**The natural accuracy of RSTs and RTTs.** As observed in Fig. 3, RSTs drawn from randomly initialized networks achieve a comparable natural accuracy with the RTTs drawn from naturally/adversarially trained networks and adversarial RTTs generally achieve the best natural accuracy.

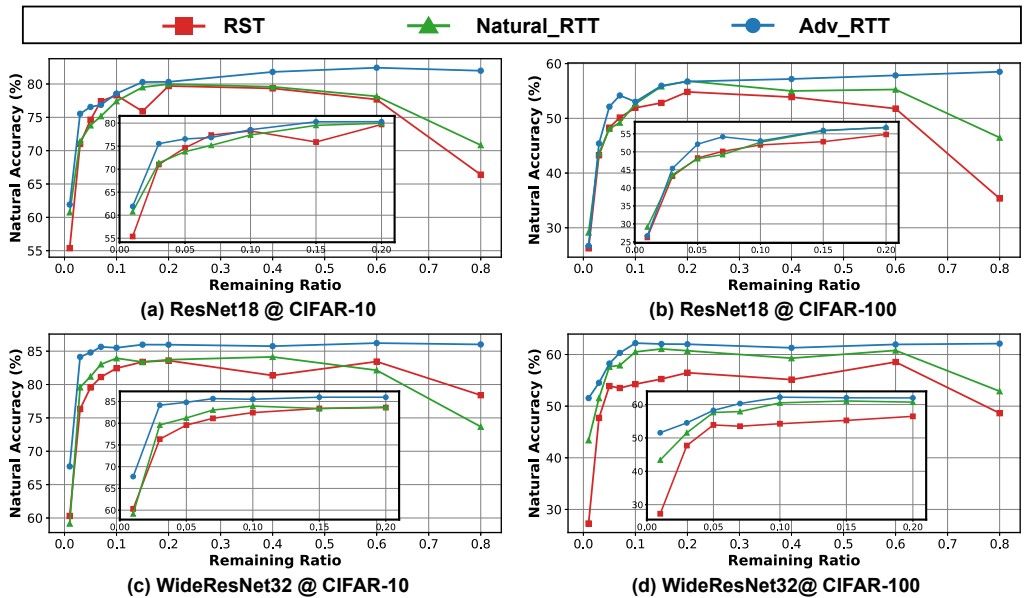

Figure 3: The natural accuracy achieved by RSTs, natural RTTs, and adversarial RTTs drawn from ResNet18/WideResNet32 on CIFAR-10/100, with zoom-ins under low remaining ratios (1%~20%).

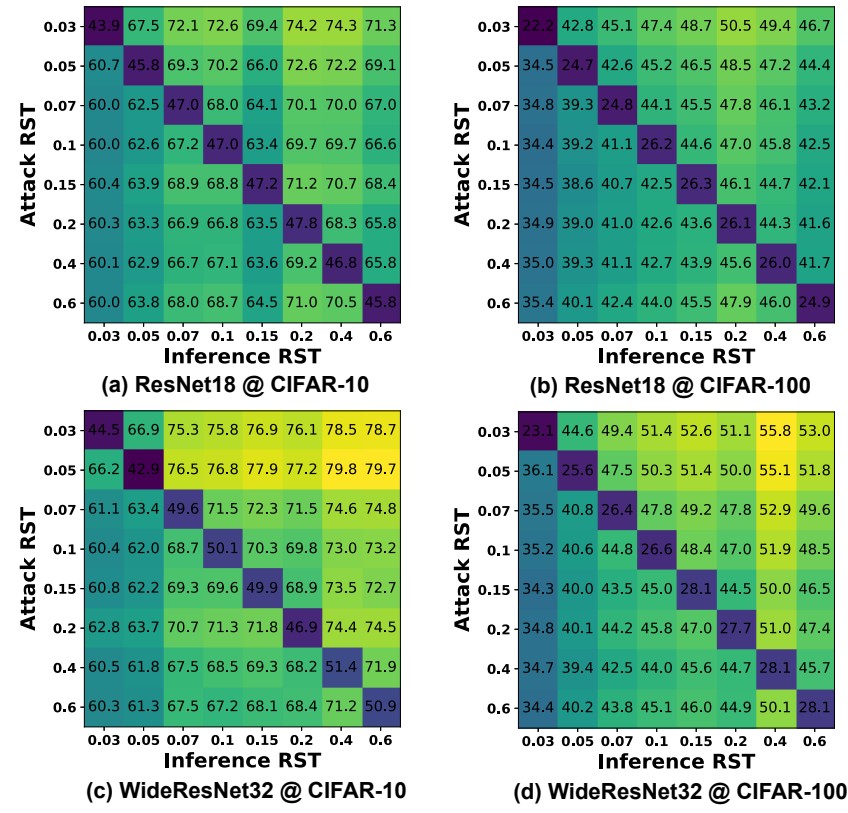

Figure 4: The adversarial transferability between RSTs with different weight remaining ratios drawn from ResNet18/WideResNet32 on CIFAR-10/100, where the robust accuracy is annotated.

## 5    More Visualizations of the Adversarial Transferability between RSTs

As shown in Fig. 4, the adversarial transferability between RSTs with different weight remaining ratios is consistently poor among all the considered networks and datasets based on the robust

Table 2: Evaluating the robustness of RSTs with different remaining ratios against the tranferred attacks generated by (1) naturally trained dense models (Dense Nat. Trained), (2) adversarially trained dense models (Dense Adv. Trained), and (3) another RST with the same remaining ratios searched from the same dense network initialized by different random seeds (Same Ratio) on top of ResNet18 and CIFAR-10.

| Attack Source | RST @3% | RST @5% | RST @7% | RST @10% | RST @15% | RST @20% | RST @40% | RST @60% | Dense Nat. Trained | Dense Adv. Trained |
|---|---|---|---|---|---|---|---|---|---|---|
| **Dense Nat. Trained** | 70.70 | 74.35 | 77.20 | 77.71 | 75.55 | 79.22 | 78.85 | 77.33 | 0 | 81.28 |
| **Dense Adv. Trained** | 60.17 | 62.01 | 65.33 | 64.92 | 62.37 | 66.78 | 66.89 | 64.93 | 81.96 | 50.19 |
| **Same Ratio** | 62.01 | 64.78 | 64.77 | 65.61 | 69.94 | 68.22 | 67.84 | 66.41 | - | - |

accuracy gap between those in the diagonal and non-diagonal elements. This indicates that the proposed R2S technique is generally applicable to different networks and datasets.

We further provide more results regarding the adversarial transferability between (1) dense models and RSTs, and (2) RSTs with the same remaining ratios searched from the same dense network initialized by different random seeds. From Tab. 2, we can observe that (1) the adversarial transferability from the naturally/adversarially trained dense models to RSTs is still poor; and (2) the robust accuracy against the attacks from RSTs of the same remaining ratio still transfer poorly, indicating that the adversaries cannot train or search another subnetwork of the same remaining ratio as an effective proxy.

Table 3: Evaluating R2S with different candidate RST sets on top of ResNet18/WideResNet32 on CIFAR-10/100. Here 'dense' denotes the adversarially trained dense networks which are the baselines.

| Dataset | CIFAR-10 | | | | CIFAR-100 | | | |
|---|---|---|---|---|---|---|---|---|
| Network | ResNet18 | | WideResNet32 | | ResNet18 | | WideResNet32 | |
| Candidate RST Set | Natural Acc (%) | PGD-20 Acc (%) | Natural Acc (%) | PGD-20 Acc (%) | Natural Acc (%) | PGD-20 Acc (%) | Natural Acc (%) | PGD-20 Acc (%) |
| Dense | 81.73 | 50.19 | 85.93 | 52.27 | 56.88 | 26.94 | 61.14 | 29.81 |
| 3%, 5%, 7%, 10% | 75.34 | 61.03 | 79.86 | 64.91 | 48.40 | 36.77 | 52.35 | 41.04 |
| 3%, 5%, 7%, 10%, 15%, 20%, 30% | 76.56 | **64.02** | 81.08 | **67.86** | 50.85 | 40.43 | 53.72 | **43.51** |
| 7%, 10%, 15%, 20%, 30% | **78.06** | 63.60 | **82.34** | 66.96 | **52.88** | 40.56 | **54.89** | 42.47 |

## 6 R2S with More Candidate RST Sets

We evaluate R2S with different candidate RST sets in Tab. 3. We can observe that (1) R2S consistently and aggressively improves the robust accuracy compared with the adversarially trained dense baselines, e.g., up to a 13.83%/15.59% and 13.62%/13.70% robust accuracy improvement on top of ResNet18/WideResNet32 on CIFAR-10 and CIFAR-100, respectively; and (2) R2S when using a candidate RST set with a wider range can generally achieve a better robust accuracy, while RSTs with weight remaining ratios that are too low or too high will influence the average natural accuracy. As such, it is suggested that the candidate RST sets are constructed within a specific range, e.g., the last row in Tab. 3.

## 7 The Adversarial Transferability between Adversarial RTTs

As shown in Fig. 5, the adversarial transferability between adversarial RTTs with different weight remaining ratios is also poor, showing a consistent phenomenon as the one between RSTs in Sec. 5. This indicate that the proposed R2S technique can be also applied on top of adversarial RTTs (here R2S denotes Random adversarial RTT Switch), which can potentially achieve better performance considering the decent robust and natural accuracy of adversarial RTTs as analyzed in Sec. 4 and Sec. 5.2 in the main text.

## 8 R2S on top of Adversarial RTTs

We apply R2S on top of adversarial RTTs as shown in Tab. 4. We can observe that it can also aggressively improve the robust accuracy, e.g., a 14.95% and 15.09% robust accuracy improvement

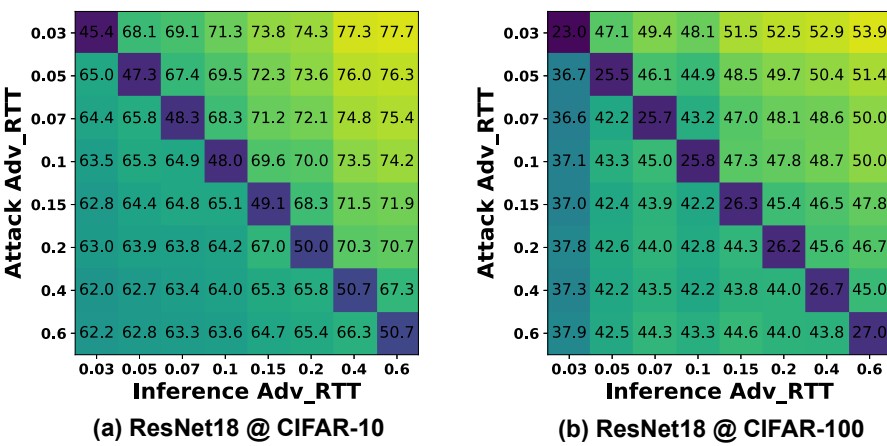

**(a) ResNet18 @ CIFAR-10**    **(b) ResNet18 @ CIFAR-100**

Figure 5: The adversarial transferability between adversarial RTTs with different weight remaining ratios in ResNet18 on CIFAR-10/100, where the robust accuracy is annotated.

Table 4: Evaluating R2S on top of adversarial RTTs on ResNet18 and CIFAR-10/100. Here 'dense' denotes the adversarially trained dense networks which are the baselines.

| Dataset | CIFAR-10 | | CIFAR-100 | |
|---|---|---|---|---|
| Candidate RTT Set | Natural Acc (%) | PGD-20 Acc (%) | Natural Acc (%) | PGD-20 Acc (%) |
| Dense | 81.73 | 50.19 | 56.88 | 26.94 |
| 3%, 5%, 7%, 10% | 77.33 | 61.97 | 51.19 | 38.73 |
| 3%, 5%, 7%, 10%, 15%, 20%, 40% | 78.57 | **65.14** | 53.52 | **42.03** |
| 7%, 10%, 15%, 20%, 40% | **79.57** | 64.15 | **55.42** | 41.38 |

over the adversarially trained dense networks with a comparable natural accuracy. This indicates that our R2S can also serve as a plug-in technique on top of existing adversarial training methods, i.e., one option is to do adversarial training first, then draw adversarial RTTs, and finally perform inference with R2S.

# 9 Advantages of RSTs over Adversarial RTTs on More Datasets

In Sec. 6.3 of the main text, we analyze that a unique property of RSTs is their advantageous parameter efficiency, i.e., the same randomly initialized network can be shared among different tasks, and task-specific RSTs can be drawn from this same network, while the task-specific adversarial RTTs drawn from adversarially trained networks on a different task will lead to a reduced robustness. We further validate this on another two datasets, i.e., CIFAR-10 and SVHN [10].

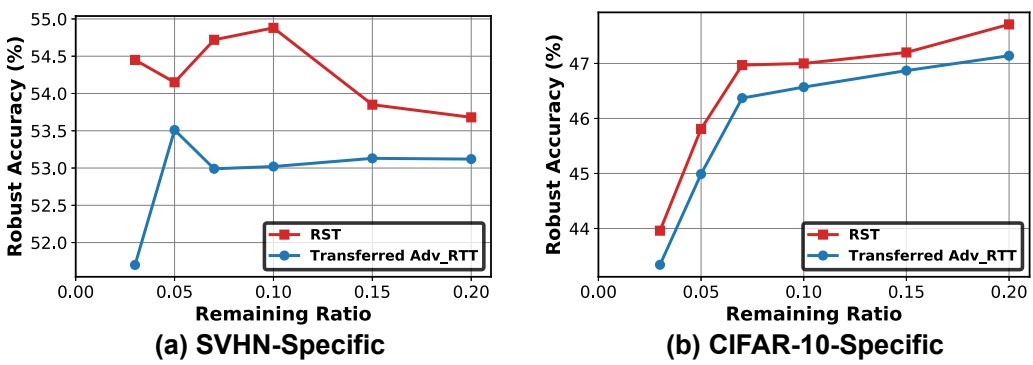

**(a) SVHN-Specific**      **(b) CIFAR-10-Specific**

Figure 6: Comparing transferred adversarial RTTs with RSTs identified in ResNet18 on (a) SVHN and (b) CIFAR-10.

**Experiment setup.** We conduct adversarial search on SVHN with the same settings as that on CIFAR-10/100 in Sec. 2 except here the initial learning rate is 0.01. We conduct two sets of comparisons: (1) SVHN-specific RSTs from ResNet18 vs. SVHN-specific adversarial RTTs drawn from ResNet18 adversarially trained on CIFAR-10 and (2) CIFAR-10-specific RSTs from ResNet18 vs. CIFAR-10-specific adversarial RTTs drawn from ResNet18 adversarially trained on SVHN. In particular, all the RSTs are drawn from ResNet18 with the same random initialization.

**Results and analysis.** As shown in Fig. 6, a consistent observation as Sec. 6.3 of the main text can be found, i.e., RSTs consistently outperform the transferred adversarial RTTs drawn from the adversarially trained network on another task. For example, RSTs achieve up to a 2.75% higher robust accuracy than transferred adversarial RTTs on SVHN under the same weight remaining ratio. This further indicates that it is a unique advantage of RSTs for being able to improve parameter efficiency, especially when more tasks are considered under resource constrained applications.