# OpenReview forum: "Drawing Robust Scratch Tickets: Subnetworks with Inborn Robustness Are Found within Randomly Initialized Networks"
_NeurIPS.cc/2021/Conference — NeurIPS 2021 Poster_

### Official Review · Reviewer_SpL6 · 2021-07-05

**Rating:** 7
**Confidence:** 4

**Summary:**

The paper describes the existence of highly robust subnetworks within neural networks with random weights. Further, the paper proposes a simple way to search for these subnetworks, characterizes their properties w.r.t. initialization, model size, attacks, models, search method, and the sparsity structure, and, finally, proposes a technique for ensembling these subnetworks into a more robust model. The experimental framework appears to be sound, and the results are impressive, showing the consistent existence (and high performance) of these subnetworks across several factors.

**Limitations And Societal Impact:**

Yes, they have.

**Main Review:**

—> Strengths:
- The paper’s main subject of study is interesting for a large portion of the DL community (and specially for the robustness community), and its main observation can spur further works that exploit the phenomenon of RSTs.

- The paper’s observation is strongly related to “What's Hidden in a Randomly Weighted Neural Network?”, Ramanujan et al., CVPR 2020 (reference [53] in the paper), and, in my opinion, is a clear contribution upon [53]’s observation: the phenomenon described in [53] also applies for robust networks by incorporating a robustness notion in the searching objective.

- The experimental framework appears to be sound, spanning a sensible amount of factors around the phenomenon being observed, which suggests the results are reliable. The results themselves are significant: the robustness of RSTs is comparable with that of adversarially trained networks.

- Overall, the paper is well written, clear, and concise. The ideas and sections are well organized, and the results are presented in a convenient manner for the reader.

—> Weaknesses that affect my rating:
- My main concern is regarding the evaluation setup presented in L335 (it is still unclear to me). I understand the motivation behind “ensembling” several models from the same network, but I think this may lead to gradient obfuscation, which should then be studied on its own. To study whether the gains in robustness come from a phenomenon beyond gradient obfuscation, then probably the attacker would have to have access to the “expected” gradient, right? So that gradients are more informative and the adversary can be properly constructed. The first thing that comes to mind is literally the mathematical definition of the gradient under expectation, but perhaps the authors could think of a more reasonable approach that would fit the definition of “adaptive” attack.

- The paper proposes a search strategy for the weights to be pruned (Sec. 3.2). The proposed strategy is sensible (based upon previous works) and the results back the choice for this strategy. However, I think it would also be interesting to study how alternative strategies compare against this one. Have the authors considered alternatives?

- While it is interesting that RSTs exist under all the initializations that were considered (Sec. 4.4), I think a similarly interesting question is what initializations would not allow RSTs to exist. That is, if we initialized with, let’s say, purely normal distribution, would that impede RSTs from existing? I do not mention this experiment as a drawback in the paper, but as a phenomenon that would be worthy of study based on the paper’s findings.

- Section 6.1 reports that the diagonal of the matrix in Fig. 7 corresponds to the lowest values observed. My understanding is that given a single randomly initialized NN one can find *various* RSTs with a given weight remaining ratio. While the figure reports that attacks transfer poorly across NNs of *different* remaining ratios, I wonder how is the transferability across NNs of the same remaining ratio (not the same network topology perhaps, but the same ratio).



—> Weaknesses that do not affect my rating but should be corrected/answered within reason:
- L68: random (or randomized?)
- L70: example
- L116: it is a bit weird to make reference to an equation that has not been introduced yet (or is not introduced immediately after).
- In Eq. (1), should not the “m” below the argmin have a hat?
- In L120 and L121, “permutation” or “perturbation”?
- In Eqn. (2), in practice, is there also a clipping for usual image pixel values, i.e. [0,1]?
- L137: sparse or sparsity?
- L145: ImageNet
- L161: decent
- Table 1 suggests a possible relation between depth (rather than parameter count) and the robustness of RSTs. Have the authors observed some phenomenon in this direction? I think there may be an interesting observation regarding depth to be made here.
- L182: “achieves” makes reference to RSTs? If so, it should be singular
- Sec. 4.6: another interesting “adversarial search method” would be simple Gaussian noise or uniform noise. It would be interesting to find whether any of these noises could be used as a reliable way of searching for RSTs.
- In Fig. 2(b) there is a regime of the “Remaining Ratio” dimension for which the fine tuning from inherited weights generates a drop in robust accuracy with respect to the original RST, i.e. the green curve is below the red curve. Is there an explanation for this phenomenon? Is this an implementation error?
- In Sec. 5.1 I assume fine tuning means “adversarial” fine tuning, and so perhaps this clarification could be useful for the reader.
- L237: achieves
- In L261, does this setup mean that the images are all added noise coming from a uniform distribution from [-eps,+eps]? Or {-eps,+eps}?
- Section 5.2, L271, studies the role of overparameterization. The conclusion, as reported in L280, is that overparameterization during adversarial training is necessary towards decent robustness of RSTs. I think I understand what the paper means, but is it not semantically incorrect to say that “training is necessary towards the robustness of Robust *Scratch* Tickets”? They are either trained or from scratch, but cannot be both, right? Or is there something I am missing?
- L275: I think there is a missing noun there, probably “accuracy”? Or “robustness” instead of “robust”?
- I think Sec 5.4 could be written in a more convenient manner: the whole section is currently a single sentence.
- L293: in the Appendix.
- L296: an.
- Fig. 7 legend: “accurracies”.
- L346: unwanted space.
- The experimental setup described in L358 is a bit unclear to me. My current understanding is: the networks is adversarial trained on CIFAR10, the last fully connected layer is randomly re-initialized, and then a search for robust networks is performed on this network. Is this correct? If so, I think the wording could improve as to better convey this setup.



**Time Spent Reviewing:**

7

---

> ### Author Response · Authors · 2021-08-10
> **Response to Reviewer SpL6**
>
> Thanks for recognizing the impact of our work to the community and for your valuable suggestions. Below are our detailed response:
>
> **1. Evaluation against adaptive attacks**
>
> Thanks for your suggestions! As advised, we design two adaptive attacks on top of PGD-20 attacks ($\epsilon=8$) to further evaluate our R2S technique: (1) EOT (Expectation over Transformation) following [1], which generates adversarial examples via the expectations of the gradients from all candidate RSTs; and (2) Ensemble, which generates attacks based on the ensemble of all the candidate RSTs, whose prediction is the averaged results of all candidate RSTs. Both attacks consider the information of all candidate RSTs and the results are summarized in the table below (the same R2S settings as Table 2 in our paper). We can observe that the robust accuracy of R2S against the two adaptive attacks will drop compared with our R2S under vanilla PGD-20 attacks, yet it’s still 7.40%~9.83% higher than the adversarially trained dense networks, indicating the two adaptive attacks are effective adaptive attacks and our R2S can still be a strong technique to boost adversarial robustness. We believe such discussions could strengthen our paper and will add more details in the final version.
>
> |     Dataset    | CIFAR-10 |   CIFAR-10   | CIFAR-100 |   CIFAR-100  |
> |:--------------:|:--------:|:------------:|:---------:|:------------:|
> |     Method     | ResNet18 | WideResNet32 |  ResNet18 | WideResNet32 |
> | Dense @ PGD-20 |   50.19  |     52.27    |   26.94   |     29.81    |
> |  **R2S @ PGD-20**  |   63.93  |     67.55    |   41.01   |     43.27    |
> |    **R2S @ EOT**   |   57.59  |     64.98    |   36.68   |     40.17    |
> | **R2S @ Ensemble** |   59.05  |     62.10    |   35.98   |     38.25    |
>
> **2. Alternative strategies for searching for RSTs**
>
> As suggested and inspired, we use gaussian noise and uniform noise, following [2] and [3] respectively, to replace our adversarial search, and find that their resulting searched RSTs show inferior robustness (with a robust accuracy of no more than 8%) against PGD-20 attacks, indicating that the current adversarial search is necessary for guaranteeing the worst-case robustness against the strong PGD-20 attacks. We will discuss potential alternatives in the final version and explore simpler proxies to search RSTs in our future work.
>
> **3. Start from other initializations**
>
> Following your inspiring question, we search for RSTs in ResNet18 initialized with (1) normal distribution and (2) uniform distribution [-1,1] on CIFAR-10, and find that the searched RSTs are inferior, e.g., achieving a 37.93%/36.39% robust accuracy with normal/uniform initializations which is much lower than the one with signed kaiming constant initialization with a robust accuracy of 46.97%. Our understanding is that the normal/uniform distributions are not able to keep the variance of each layer to be approximately constant and thus a small perturbation applied to the inputs can lead to a large change in the following features, leading to the difficulty to find RSTs with both decent natural accuracy and robustness against perturbations. Therefore, we suggest starting with commonly adopted deep neural network initializations, and agree that initializations impeding RSTs are worthy of study, for which we will add more discussions in the final version.
>
> **4. The transferability across NNs of the same remaining ratio**
>
> Following your curiosity, we start from different random seeds (under the same type of initialization) and show the robust accuracy against transferred attacks from NNs of the same remaining ratio in the table below. We can see that the adversarial transferability is still poor, indicating that the adversaries cannot use another NN of the same remaining ratio as an effective proxy.
>
> |      Model     | RST 0.03 | RST 0.05 | RST 0.07 | RST 0.1 | RST 0.15 | RST 0.2 | RST 0.4 | RST 0.6 |
> |:--------------:|:--------:|:--------:|:--------:|:-------:|:--------:|:-------:|:-------:|:-------:|
> | Robust Acc (%) |   62.01  |   64.78  |   64.77  |  65.61  |   69.94  |  68.22  |  67.84  |  66.41  |
>
> **5. Relation between depth/width and the robustness of RSTs**
>
> To study the influence of depth/width on RSTs’ robustness, we increase the depth of ResNet18 by 2x/3x/4x and compare it with their corresponding wider variants (i.e., with channel numbers being scaled up) under the same FLOPs. We find that RSTs drawn from wider networks are consistently more robust (with 1~2% higher robust accuracy under the same pruning ratio), aligning with the common belief that wider networks are more robust than deeper ones. We will add more discussions in the final version.
>
> **6. Explanation for Fig 2(b)**
>
> We find that the fine-tuned RSTs with inherited weights (the green curve in Fig 2(b)) at a remaining ratio of 0.08 is due to the training variance, as the green curve will be smooth at this point when averaged over more runs (i.e., five instead of three). Currently the results are averaged over three runs and we will further include more runs in the final version.
>
> **7. The claim for the role of overparameterization**
>
> Sorry for the confusion. Here we want to highlight that overparameterization during adversarial training is necessary towards decent robust subnetworks. We will clarify this in the final version.
>
> **8. Experiment setup in L358**
>
> Yes you are right. Thanks for pointing out and we will improve the wording for this part.
>
> **9. Typos and formatting issues**
>
> Thanks for pointing out the typos and formatting issues! We will correct them and perform more careful proofreading for the final version.
>
> **10. Other questions**
>
> Thanks for your careful reading/catch! We will further clarify in the final version. In Eq.(1), m below the argmin does not have a hat, indicating that all the elements in m will be updated during the backward pass via straight-through estimation. In L120/121, it should be "perturbation". In Eq.(2), there’s also a [0,1] clipping for $x_i+\delta_i$ by default and we omit that for simplicity. In L137, it should be "sparsity". In L182, it should be “achieve”. In Sec. 5.1, it should be "adversarial fine tuning" and we will clarify in the final version. In L261, it should be {-eps, eps}. In L275, an “accuracy” is missing here.
>
> - **Reference**
>
> [1] “On Adaptive Attacks to Adversarial Example Defenses”, F. Tramer et al., NeurIPS’20
>
> [2] “Certified Adversarial Robustness via Randomized Smoothing”, J. Cohen et al., ICML’19
>
> [3] “Certified Adversarial Robustness with Additive Noise”, B. Li et al., NeurIPS’19

---

> > ### Comment · Reviewer_SpL6 · 2021-08-29
> > **Response to authors**
> >
> > Thanks for conducting further experiments.
> >
> > I particularly value the experiments on the adaptive attacks. I think these should appear in the paper for completeness.
> >
> > Looking into what the other reviewers mentioned, I agree with them that the paper could strongly benefit from adding discussion to the previous works mentioned in the other reviews. I expected that adding this discussion will clarify the paper's contributions to the readers.
> >
> > Taking all this into consideration, I maintain my decision of accepting the paper. I think its results are experimentally sound, the observations are worthwhile, and that there can be interesting potential applications.

---

> > > ### Author Response · Authors · 2021-08-30
> > > **Follow up**
> > >
> > > Thank you for recognizing the potential impacts of our work and for your constructive comments which would help improve our paper! We will follow your and other reviewers’ suggestions to include the discussions about adaptive attacks and the comparison with other methods in the final version.

---

### Official Review · Reviewer_pV21 · 2021-07-07

**Rating:** 6
**Confidence:** 5

**Summary:**

This paper discovers the existence of robust subnetworks coined as Robust Search Tickets (RSTs) within randomly initialized networks that attain comparable or better performance than the adversarial trained networks. The paper conducts extensive experiments to study the existence of RSTs under different models, datasets and sparsity patterns. Additionally, the paper identifies the poor transferability between RSTs of varying sparsity ratios. To tackle this problem, it proposes a novel method called Random RST Switch (R2S) that randomly switches between different RSTs.

**Limitations And Societal Impact:**

Yes

**Main Review:**

## Strengths

* As network pruning is getting adopted in resource-constrained applications, the lottery ticket hypothesis is one approach that has been found effective in over-parameterized deep neural networks. Thus, the paper is well-motivated and showcases an interesting phenomenon of robust search tickets within randomly initialized networks.
* The results are intriguing, especially the comparison between RST and Robust Trained Tickets (RTTs) and the investigation of adversarial transferability between RSTs are interesting.
* The paper is clearly written, easy to follow and provides interesting insights into robust tickets.

***

## Weaknesses
* **Significance and novelty:** One of my primary concerns is the novelty of the work. Ye et al. [1] discuss the lottery ticket hypothesis in the adversarial setting (see Section 4.2). Cosentino and Zaiter et al. [2] analyzed Lottery Ticket Hypothesis with adversarial training. Li and Wang et al. [3] also demonstrate the winning tickets and boosting tickets exist for adversarial training. The paper should compare and highlight the differences with these works.
* **Comparison with existing works:**
     * The paper highlights existing works that utilize pruning for adversarial robustness but do not provide an empirical comparison with those methods, which makes it challenging to evaluate the efficacy of the proposed method for robustness. I would suggest having the experimental comparison with Ye et al. [1], Madaan et al. [4] and Sehwag et al.[5] to confirm the fact that the proposed method works better than the existing techniques for robustness using pruning.
    * Additionally, while the proposed method shows slight improvement, Madry et al. is the strongest considered baseline. I would recommend including more robust baselines following RobustBench [6] to highlight the effectiveness of the proposed method over the dense baselines.
* **Limited comparison:** The comparison with dense networks is unfair for Imagenet, in my opinion. For instance, in Table 1, the paper compares ResNet-50/ResNet-110 for RST while using ResNet-18 for the Dense Network. Instead, for a fair comparison, the paper should use the same base network. Furthermore, the paper should have used AutoAttack for all the evaluation results instead of using PGD-20, which is a significantly weaker attack in comparison.

***

## Other Questions and Comments
* RST on Auto Attack does not show improvement over the dense network in Table 1 in the Appendix. Comment.
* It is well known that adversarially trained models overfit to single $\ell_p$ norm. Since RST does not use adversarial training, can it generalize to multiple $\ell_p$ norms and attacks? I believe including these results can further strengthen the paper.
* In Figure 7, please include the transferability from a dense standard, dense adversarial trained model to RST and dense to dense for comparison.
* The proposed acronym RST also conflicts with Robust Self-Training [7] in the robustness literature. The acronym should be altered to avoid confusion in future work.
* The code and the pre-trained models are not provided with the paper or supplementary material. I would encourage the authors to release the code to promote reproducibility and successful comparison for future works.
* I would also suggest the authors report the variance for the Tabular results; it is also good practice for the community.

***
## References

[1] Zhang et al. Prune DNN using Alternating Direction Method of Multipliers (ADMM). ECCV 2020

[2] Cosentino and Zaiter et al. The Search for Sparse, Robust Neural Networks.

[3] Li and Wang et al. Towards Practical Lottery Ticket Hypothesis for Adversarial Training.

[4] Madaan et al. Adversarial Neural Pruning with Latent Vulnerability Suppression. ICML 2020

[5] Sehwag et al. HYDRA: Pruning Adversarially Robust Neural Networks. NeurIPS 2020

[6] Croce, Andriushchenko, Sehwag and Debenedetti et al. RobustBench: a standardized adversarial robustness benchmark.

[7] Carmon et al. Unlabeled Data Improves Adversarial Robustness. NeurIPS 2019

**Time Spent Reviewing:**

10

---

> ### Author Response · Authors · 2021-08-10
> **Response to Reviewer pV21**
>
> Thanks for recognizing the insights of our work to the community and for your valuable suggestions. Below are our detailed response:
>
> **1. Novelty of this work**
>
> We humbly clarify that our RST is fundamentally and principally different from all the previous adversarial lottery ticket works you mentioned: we are the first to identify subnetworks with inborn robustness in randomly initialized networks, i.e., **neither the original dense networks nor the identified RSTs need to be trained**, as illustrated in our abstract, introduction, and Section 2 (line 90~98). It is worth noting that the drawn RSTs with inherited weights from the randomly initialized networks are robust without being trained in isolution, which is yet required by the mentioned previous lottery ticket works. As such, our work is a fundamentally new finding as recognized by the other reviewers commenting as “intriguing and underexplored” and “interesting for a large portion of the DL community”. Furthermore, we conduct an extensive study about the existence and properties of RSTs and propose the R2S technique on top of that to practically utilize RSTs to boost robustness.
>
> **2. Comparison with other adversarial lottery ticket works**
>
> Following your suggestion, we benchmark with [2] since [1] and [3] only focus on the small-scaled MNIST, and find that [2] achieves 50.48% robust accuracy with a sparsity of 80% on WideResNet32/CIFAR-10 under PGD-20 attacks with $\epsilon=8$, while our RST achieves 51.39% robust accuracy with a sparsity of 90%. Additionally, we show that when being adversarially fine-tuned (i.e., fine-tuned RSTs) or drawn from adversarially trained dense networks (i.e., adversarial RTTs), the robustness of our RSTs can be further improved in Section 5.1 and 5.2. As suggested, we will add more comparison with [1-3] and further clarify that the settings and context of RSTs are different in the final version.
>
> **3. Comparison with pruning baselines**
>
> We humbly clarify that our work does not propose a new pruning technique, instead we for the first time discover the existence of RSTs with inborn robustness without training the original dense networks or RSTs and thoroughly study their properties, on top of which the R2S technique is proposed to win both robustness and efficiency. Meanwhile, as suggested, here we compare R2S with the settings in Table 2 of our paper (14.5% average pruning ratio) with the suggested adversarial pruning baselines and find that our R2S, achieving a 64.0% robust accuracy on VGG16/CIFAR-10 under PGD-20 attacks with $\epsilon$=8, outperforms all the baselines in [4][5] under the same setting.
>
> **4. Comparisons with more baselines in RobustBench**
>
> Following your suggestion, we further compare R2S with the SOTA top 6 solutions (B1~B6 are listed below) for WideResNet-34-10/CIFAR-10 (without resorting to extra data) under AutoAttack with $\epsilon=8$ in RobustBench as shown in the following table. We can see that R2S still achieves the most competitive performance with comparable or better robustness. It is worth noting that many of the SOTA solutions are orthogonal with our R2S and can be integrated into our adversarial search process for better RSTs. In addition, our R2S has two additional advantages: (1) the average sparsity of R2S is only 14.5% while all the SOTA methods are dense models; (2) task-specific RSTs can be drawn from the same randomly initialized network and stored compactly via instantiated binary masks according to Section 6.3.
>
> |   Method   | Natural Acc (%) | Robust Acc (%) |
> |:----------:|:---------------:|:--------------:|
> |     B1     |      85.36      |      56.17     |
> |     B2     |      84.52      |      53.51     |
> |     B3     |      83.48      |      53.34     |
> |     B4     |      84.92      |      53.08     |
> |     B5     |      88.22      |      52.86     |
> |     B6     |      85.32      |      51.12     |
> | **R2S (ours)** |    **81.87**      |      **56.06**     |
>
> - **The top 6 baselines from RobustBench:**
>
> B1: “Adversarial Weight Perturbation Helps Robust Generalization”, D. Wu et al., NeurIPS’20
>
> B2: “Attacks Which Do Not Kill Training Make Adversarial Learning Stronger”, J. Zhang et al., ICML’20
>
> B3: “Self-Adaptive Training: beyond Empirical Risk Minimization”, L. Huang et al., NeurIPS’20
>
> B4: “Theoretically Principled Trade-off between Robustness and Accuracy”, H. Zhang et al., ICML’19
>
> B5: “Learnable Boundary Guided Adversarial Training”, J. Cui et al., ICCV’21
>
> B6: “Efficient Robust Training via Backward Smoothing”, J. Chen et al., arXiv’20
>
> **5. Comparison with dense networks on ImageNet**
>
> We humbly clarify that our key insight here is that RSTs, hidden within randomly initialized networks with more overparameterization, achieve better robustness than the adversarially trained dense networks with a similar model size, as illustrated in Section 4.2 (line 166) and 4.3 (line 184), following the practice of [53] (Section 4.6 and Figure 8).
>
> **6. Evaluation results against AutoAttack**
>
> We show the evaluation results against AutoAttack in Table 3 for R2S and Appendix Table 1 for RSTs. We mainly use PGD-20 attacks since (1) it's more time-efficient, and (2) it still serves as one major evaluation criteria in the latest ICLR’21 and ICML’21 works. Thanks for the suggestion and we will include more ablation studies using AutoAttack in the final version.
>
> **7. RST on Auto Attack does not show improvement over the dense network**
>
> We want to clarify that we never claim RSTs can surpass the adversarially trained dense networks, which is also difficult for all adversarial pruning methods under a large prune ratio. Here RSTs with only 5% remaining weights (i.e., 95% sparsity) can achieve robustness very close (-0.76%~+0.30%) to  the dense ones under AutoAttack, indicating the general existence of RSTs under various attacks. In addition, the proposed R2S technique aims at improving the robustness over their corresponding dense networks and the evaluation against AutoAttack can be found in Table 3.
>
> **8. Generalize to multiple $L_{p}$-norm attacks**
>
> Thanks for your constructive suggestion! We further evaluate RSTs with different remaining ratios, R2S with the same setting in Table 2, and their corresponding dense networks (trained by $L_{inf}$-PGD) against $L_{2}$-PGD attacks on top of ResNet18/WideResNet32 on CIFAR-10. As shown in the table below, we can see that with the increased $\epsilon$, the dense networks suffer from larger robust accuracy drops and RSTs can gradually outperform the dense networks by a notable margin, indicating that RSTs will suffer less from the overfitting to a specific $L_{p}$-norm than the dense networks. In addition, R2S consistently achieves the best results here. We will add more results and discussion in this regard.
>
> |        Model        |          | ResNet18 |          |          | WideResNet32 |          |
> |:-------------------:|:--------:|:--------:|:--------:|:--------:|:------------:|:--------:|
> |        Method       | $\epsilon$=0.50 | $\epsilon$=1.71 | $\epsilon$=2.00 | $\epsilon$=0.50 |   $\epsilon$=1.71   | $\epsilon$=2.00 |
> |        Dense        |   63.61  |   40.85  |   39.04  |   63.94  |     36.55    |   34.66  |
> |        **RST 5%**       |   61.11  |   43.81  |   42.25  |   65.96  |     47.75    |   45.96  |
> |        **RST 7%**       |   63.04  |   43.94  |   42.41  |   65.05  |     43.63    |   42.08  |
> |       **RST 10%**       |   63.61  |   44.49  |   42.81  |   66.93  |     45.56    |   43.87  |
> |       **RST 15%**       |   62.03  |   44.16  |   42.58  |   67.18  |     45.23    |   43.44  |
> |       **RST 20%**       |   64.43  |   44.35  |   42.51  |   67.54  |     45.21    |   43.51  |
> | **R2S** |   66.09  |   50.07  |   48.47  |   70.35  |     53.30    |   50.93  |
>
> **9. More adversarial transferability results**
>
> Following your suggestion, we list more adversarial transferability results between dense-to-RSTs and dense-to-dense on ResNet18/CIFAR-10 under transferred PGD-20 attacks in the table below and observe that the adversarial transferability from the dense standard/adversarially trained models to RSTs is still poor. We will add more results and discussion in this regard.
>
> |   Attack Source   | RST 3% | RST 5% | RST 7% | RST 10% | RST 15% | RST 20% | RST 40% | RST 60% | Dense std trained | Dense adv trained |
> |:-----------------:|:------:|:------:|:------:|:-------:|:-------:|:-------:|:-------:|:-------:|:-----------------:|:-----------------:|
> | Dense std trained |  70.7  |  74.35 |  77.2  |  77.71  |  75.55  |  79.22  |  78.85  |  77.33  |         0         |       81.28       |
> | Dense adv trained |  60.17 |  62.01 |  65.33 |  64.92  |  62.37  |  66.78  |  66.89  |  64.93  |       81.96       |       50.19       |
>
>
> **10. The acronym of RST**
>
> Thanks for the suggestion! We will use a better acronym in the final version.
>
>  **11. Open source code**
>
> As promised in the abstract, we will release all the source codes and pretrained models upon acceptance.
>
> **12. Variance for the tabular results**
>
> Thanks for the suggestions! We find that the variances of accuracy are generally within ±0.5% in Tables 1~3, and will add the detailed results in the final version.

---

> > ### Comment · Reviewer_pV21 · 2021-08-17
> > **Thank you for the response**
> >
> > Thank you for your answers and additional results. Here are some direct answers related to your response:
> >
> > > Comparison with other adversarial lottery ticket works
> >
> > Thank you for taking the time to conduct these experiments. I believe including the comparison and discussion with these baselines should further strengthen the contributions of the paper.
> >
> > > Comparison with pruning baselines
> >
> >  I realize that the paper shows the existence of robust search tickets within randomly initialized networks. However, I still believe the ideal objective is to obtain robust and sparse networks; therefore, it is crucial to compare existing pruning methods that are robust and R2S on a comparable scale to provide useful insights for the community. Thank you for doing these comparisons. I appreciate it and would suggest including them in the final revision.
> >
> > > Comparisons with more baselines in RobustBench
> >
> > Thank you for adding these comparisons. Great new results.
> >
> > >  Evaluation results against AutoAttack
> >
> > I can't entirely agree with the comment of using PGD-20 attacks as a major evaluation criterion in the latest ICLR'21 and ICML'21 works. I would refer the authors to Tramer et al. [8], which broke 13 defenses using adaptive attacks. Consequently, it is necessary to thoroughly evaluate using adaptive attacks to prevent obfuscated gradients.
> >
> > > Generalize to multiple-norm attacks
> >
> > Thank you for these results. I believe the inclusion of generalization to "unseen" attacks will be valuable for the readers.
> >
> > > More adversarial transferability results
> >
> >  Thank you for trying this out.
> >
> > ---
> >
> > Overall, thank you for conducting the additional experiments. The new results demonstrate interesting insights and the utility of the proposed method over the existing methods. The authors' response addressed some of my concerns, and I adjusted my rating accordingly towards acceptance. I hope the authors can address the remaining concerns in the paper update and include the provided results with the discussion in the final revision.
> >
> > References
> > [8] Tramer et al. On Adaptive Attacksto Adversarial Example Defenses. NeurIPS 2020

---

> > > ### Author Response · Authors · 2021-08-30
> > > **Follow up**
> > >
> > > Thanks a lot for your detailed response and for recognizing the interesting insights of our work! We will address your further suggestion and include the provided results with the discussion in the final revision.

---

### Official Review · Reviewer_5CcE · 2021-07-16

**Rating:** 7
**Confidence:** 4

**Summary:**

This paper studies the inherent adversarial robustness of subnetowrks inside a full network. It names such subnetworks as Robust Scratch Tickets (RSTs). RSTs have good properties, as they possess robustness even without adversarial training. The paper further studies the existence and properties of RSTs under various aspects. The overall findings are interesting and motivating regarding the architectural robustness.

**Limitations And Societal Impact:**

Yes

**Main Review:**

Strengths:
+ The analysis of robust subnetworks without training is intriguing and underexplored
+ The proposed method for finding RSTs is well-motivated
+ The experiments are thorough and cover many aspects that are related to robustness
+ The writing is clear and easy to follow

Weaknesses:
- Although the experiments thoroughly evaluated the properties of RSTs, it still remains unclear what makes RSTs robust. For example, what architecture patterns do RSTs exhibit? Is there a pattern that correlates well with network robustness? The answers could further shed light on how to design robust networks from scratch.
- Following the topic on robustness of model architecture, the authors might want to discuss the relationship to recent works on searching robust models with neural architecture search [1, 2]. Especially as indicated in [2], the designed model has much less number of parameters compared to ResNet / MobileNet-V2, but exhibits higher robustness. It would be interesting to also extend experiments to analyze the connections/pattern similarities of RSTs and adversarially searched robust architectures.

Reference:
[1] Intriguing Properties of Adversarial Examples. 2017.
[2] When NAS Meets Robustness: In Search of Robust Architectures against Adversarial Attacks. CVPR 2020.

**Time Spent Reviewing:**

4.5

---

> ### Author Response · Authors · 2021-08-10
> **Response to Reviewer 5CcE**
>
> Thanks for appreciating the potential impacts of our work and for your constructive suggestions. Below are our detailed response:
>
> **1. Reasons behind the robustness of RSTs**
>
> The robustness of RSTs is attributed to RSTs’ searching process which ensures the searched subnetworks to effectively extract features for good natural accuracy and to identify critical weight locations for model robustness. The latter has been motivated by [1] that the location of weights holds most of the information encoded by the training, indicating that searching for the locations of a subset of weights within a randomly initialized network might be potentially as effective as adversarially training the weight values, in terms of generating robust models. We will add more such discussions in the final version and explore the theoretical support for RSTs as an exciting future work.
>
> **2. Architecture patterns of RSTs and their connections with recent robust NAS works**
>
> Although we adopt unstructured pruning in a uniform manner (i.e., the same pruning ratio for all layers) in most experiments since RSTs under unstructured pruning achieve the best robustness (see Section 4.5), we surprisingly find that the final RSTs’ sparsity resulting from an unstructured pruning scheme shows some structured patterns in the kernel/channel granularity, especially in the early layers of ResNet18/WideResNet32 for low-level feature extraction, which are more susceptible to attacks as observed in [2]. In particular, under a high pruning ratio (i.e., a low remaining ratio), the remaining weights tend to reside in the same kernel and channel, instead of uniformly distributed in the whole weight tensor. Your suggested work [3] shows that densely connected patterns result in improved robustness and our observation seems to echo this point that the clustered weights in the same kernel/channel can be viewed as locally dense connections in a region, which achieves better robustness compared with the uniformly distributed sparser patterns under the same pruning ratio. From another perspective, the uniformly distributed sparse patterns might be viewed as a series of low-rank and ill-posed weight matrices, which are inferior in robustness compared with our searched ones. We will add more discussions in the final version.
>
> - **Reference**
>
> [1] “Train-by-Reconnect: Decoupling Locations of Weights from their Values”, Y. Qiu et al., NeurIPS’20
>
> [2] “Identifying Layers Susceptible to Adversarial Attacks”, S. Siddiqui et al., arXiv’21
>
> [3] “When NAS Meets Robustness: In Search of Robust Architectures against Adversarial Attacks”, M. Guo et al., CVPR’20

---

### Decision · Program_Chairs · 2021-09-27

**Decision:**

Accept (Poster)

**Comment:**

This paper investigates whether subnetworks with high adversarial robustness can be found by learning a mask over the weights in a randomly initialized large network. Using this approach, the authors demonstrate that robust scratch tickets (RSTs) can be found for a number of datasets and show that multiple RSTs can be ensembled to further increase robustness. All reviewers found the work to be interesting, original, and of high quality, especially after a robust discussion with the authors, and I think this paper will be useful for the broader community. I would strongly encourage the authors to include all the experiments performed for the rebuttal in the final paper. I recommend acceptance.